# The Learning Curve of Urodynamics for the Evaluation of Lower Urinary Tract Symptoms

**DOI:** 10.3390/medicina58030341

**Published:** 2022-02-23

**Authors:** Matteo Frigerio, Marta Barba, Alice Cola, Silvia Volontè, Giuseppe Marino, Luca Regusci, Paola Sorice, Giovanni Ruggeri, Fabiana Castronovo, Maurizio Serati, Marco Torella, Andrea Braga

**Affiliations:** 1Department of Obstetrics and Gynecology, ASST Monza, San Gerardo Hospital, 20900 Monza, Italy; m.barba8792@gmail.com (M.B.); a.cola1@gmail.com (A.C.); s.volonte6@campus.unimib.it (S.V.); g.marino38@campus.unimib.it (G.M.); 2Department of Obstetrics and Gynecology, EOC—Beata Vergine Hospital, 6850 Mendrisio, Switzerland; luca.regusci@eoc.ch (L.R.); giovanni.ruggeri22@gmail.com (G.R.); fabiana.castronovo@gmail.com (F.C.); andrea.braga@eoc.ch (A.B.); 3Department of Obstetrics and Gynecology, G. Fornaroli Hospital, 20013 Magenta, Italy; paodotto@gmail.com; 4Department of Obstetrics and Gynecology, Insubria University, 21100 Varese, Italy; mauserati@hotmail.com; 5Obstetrics and Gynecology Unit, Department of Woman, Child and General and Specialized Surgery, University of Campania “Luigi Vanvitelli”, 80100 Naples, Italy; marcotorella@iol.it

**Keywords:** pelvic floor disorders, learning curve, urodynamics, resident, pressure/flow study, uroflowmetry

## Abstract

*Background and Objectives:* Urodynamics is considered the gold standard for lower urinary tract functional assessment. However, it requires very specific skills and training, which are currently difficult to master due to its reduced use. Moreover, no studies or data are available to define the workload and the learning curve of this diagnostic tool. As a consequence, we aimed to evaluate the learning curve of residents with no previous experience to correctly perform and interpret urodynamics, and properly address and manage patients with pelvic floor disorders based on urodynamics findings. *Materials and Methods:* This prospective study analyzed a series of proficiency parameters in residents performing urodynamics under consultant supervision, including the following: duration of procedure, perceived difficulty, need for consultant intervention, accuracy of interpretation, and therapeutic proposal. The number of procedures performed was then divided into groups of five to evaluate the progressive grade of autonomy (technical and full management autonomy) reached by each resident. *Results:* In total, 69 patients underwent urodynamics performed by three residents, with every resident performing at least 20 exams. Duration of procedure, perceived difficulty, need for consultant intervention, accuracy of interpretation, and the appropriateness of the hypothetical proposal of management/treatment based on their interpretation of clinical data and urodynamic findings was shown to be directly related to the number of exams performed. Technical autonomy in the execution of uroflowmetry was reached in the group performing 6–10 procedures, while technical autonomy in the execution of cystomanometry with pressure/flow study was obtained in the group of 16–20 procedures. The latter corresponded also to the gain of full autonomy which also included an optimal therapeutic proposal. *Conclusion:* We found that there is a tangible learning curve for urodynamics in terms of several proficiency parameters. A workload of 5 uroflowmetries and 15 cystomanometries with pressure/flow studies may be adequate to complete the learning curve.

## 1. Introduction

Pelvic floor disorders (PFDs) represent highly prevalent health problems that seriously affect patients’ quality of life [1]. These are thought to be related to pelvic floor weakening and/or tears mostly related to obstetric trauma [2]. Among PFDs, lower urinary tract symptoms (LUTS) represent the ones mostly reported, and their prevalence varies according to age, BMI, and parity [3]. However, there is great variability in instrumental findings among patients with similar clinical presentations, and this clearly may affect the type and effectiveness of management. This led to the well-known aphorism that “the bladder is an unreliable witness of itself”. Urodynamics probably represents the most important diagnostic tool for the assessment of lower urinary tract dysfunctions, as it offers a worthwhile picture of bladder functioning. Despite its usefulness, the use of urodynamics is currently under debate, due to its invasiveness and costs. Another criticism posed against urodynamics is that it requires very specific skills and training, which are currently difficult to master due to its reduced use. However, to date, there are no studies or data to help define the workload and/or the learning curve of this diagnostic tool. Learning curves represent complex functions, with the following phases: (1) an initial curve, where there is generally a stepwise improvement in learning; (2) a slower learning growth when an operator becomes more competent at a skill; (3) an expert plateau, which does not necessarily indicate an expert level; (4) after the plateau has been reached there is usually a slight decline in performance, which may be related to overconfidence and/or the ascertainment of more difficult operations [4,5].

As a consequence, we aimed to evaluate the learning curve of residents with no previous experience to correctly perform and interpret urodynamics, and properly address and manage patient PFD based on urodynamics findings. As primary outcome, we evaluated the impact of experience on the following proficiency parameters: duration of the procedure, perceived difficulty, need for consultant intervention, appropriateness of interpretation, and therapeutic proposal. As a secondary outcome, we wished to establish the workload necessary to achieve a satisfactory capability to perform urodynamics.

## 2. Materials and Methods

This was a prospective study. We analyzed all consecutive women who underwent urodynamics for LUTS performed by residents under single consultant (MF) supervision, between January and December 2020. No inclusion or exclusion criteria were applied. The residents were heterogeneous with respect to years of residency, but none of them had previously performed urodynamics. All residents voluntarily agreed to participate in the study. Before starting to perform it, they familiarized themselves with the urodynamic diagnostic tool by following a set of theoretical lessons and by viewing videos of consultant gynecologists performing the exam and then assisting the consultant in performing urodynamics. Clinical assessment was performed before urodynamics, including a medical interview to collect clinical history and lower urinary tract symptoms, and define the indication to perform the urodynamic evaluation. Patients were screened for urinary tract infection with a negative urine culture. Procedures were performed in an outpatient setting by residents, with strict mentor supervision. The urodynamic evaluation included uroflowmetry and cystomanometry with pressure/flow study as previously described [6]. All procedures and definitions conformed to the Good Urodynamic Practice Guidelines of the International Continence Society [7]. The duration of procedure for both uroflowmetry and cystomanometry with pressure/flow study were rounded up to the next 5 min and noted. The operator was asked to evaluate each exam’s perceived difficulty on a 10-point VAS scale (0 = very easy, 10 = very difficult). The need to request consultant intervention for technical issues was also noted. Accuracy of the interpretation of the exam made by the resident was evaluated by the consultant on a 10-point VAS scale (0 = very inaccurate, 10 = very accurate). Lastly, the resident was asked to make a hypothetical proposal of management or treatment based on their interpretation of clinical data and urodynamic findings, and this was rated by the consultant on a 10-point VAS scale (0 = very inappropriate, 10 = very appropriate). The characteristics of the exams (duration of procedure, perceived difficulty, need for consultant intervention, accuracy of interpretation, and therapeutic proposal) were considered as markers to evaluate learning curves for each resident. The number of procedures performed was then divided into groups of five (1–5; 6–10; 11–15; 16–20) to evaluate the progressive grade of autonomy reached by each resident. Specifically, technical autonomy in performing the exams was defined as consecutive lack of need of the consultant intervention plus optimal (≥8) accuracy of interpretation in all the exams of the group. Full autonomy was defined when technical autonomy in both exams was reached (uroflowmetry and cystomanometry with pressure/flow study) plus optimal (≥8) appropriateness of therapeutic proposal in all the exams of the group.

As this was an observational analysis, and clinical management of patients was not modified by the study, it was considered exempt from Institutional Review Board (IRB) approval from the local Ethics Committee. The study was conducted in accordance with the Declaration of Helsinki. Written informed consent was obtained from all the patients before the procedure, as part of our protocol for urodynamics. Statistical analysis was performed using JMP version 9 (SAS, Cary, NC, USA). Data were reported as mean ± standard deviation for continuous variables and as absolute frequency for non-continuous ones. Trends over time in the variables analyzed were graphically interpolated by the number of procedures performed using linear lines. The analysis of variance F test was performed, and F < 0.05 was considered statistically significant.

## 3. Results

In the period of interest, 69 patients underwent urodynamics performed by three residents. Every resident performed at least 20 urodynamics. The indication to perform urodynamics were stress urinary incontinence in 13 (18.8%) patients, urge urinary incontinence in 9 (13.0%) patients, voiding symptoms in 12 (17.4%) patients, and a combination of the previous in the remaining 35 (50.7%) patients. Duration of procedure (F ranging from <0.001 to 0.024), perceived difficulty (F ranging from <0.001 to 0.018), need for consultant intervention (F ranging from 0.001 to 0.0018), accuracy of interpretation (F ranging from <0.001 to 0.010) resulted in being statistically associated with the number of procedures executed for both uroflowmetry (Figure 1) and cystomanometry with pressure/flow study (Figure 2) for all the three residents. Moreover, the appropriateness of the hypothetical proposal of management/treatment based on their interpretation of clinical data and urodynamic findings resulted in being directly related to the number of exams performed (F < 0.001; Figure 3). The proficiency parameters for each resident were considered in groups of five consecutive procedures to evaluate the impact of experience (Table 1). Technical autonomy in the execution of uroflowmetry was reached in the group of 6–10 procedures, while technical autonomy in the execution of cystomanometry with pressure/flow study was obtained in the group of 16–20 procedures. The latter corresponded also to the gain of full autonomy which also included the optimal therapeutic proposal.

## 4. Discussion

With this paper, we aimed to evaluate the learning curve of residents with no previous experience to correctly perform and interpret urodynamics, properly address and manage patient PFD based on urodynamics findings, and define the workload necessary to achieve a satisfactory capability to perform urodynamics. We found that there is a tangible learning curve for urodynamics in terms of all considered proficiency parameters (procedure operative time, perceived difficulty, call for help, accuracy of interpretation, and appropriateness of therapeutic proposal). The learning curve for the optimal execution and interpretation of uroflowmetry corresponds to 5 procedures, while cystomanometry with pressure/flow study requires 15 procedures in order to be performed accurately. The latter also corresponds to the workload necessary to achieve accuracy while formulating a therapeutic proposal based on clinical and urodynamic findings. Interestingly, this seems to be well reproducible, since results were similar for all considered residents.

Many factors contribute to whether a physician is able to perform a particular medical procedure, including the extent of specific training and skills, which can be considered as an extension of the concept of experience. Recently, there has been great interest in the evaluation of the impact of the degree of experience in terms of the proficiency of medical procedures [8,9]. Learning curves graphically meet this need by representing the impacts of the repetitive task over a defined period, in terms of proficiency parameters, such as duration of procedure, difficulty and grade of autonomy. The concept of a learning curve is particularly useful to evaluate the trainees’ performance and consider actions such as further education, retraining, or implementation of training strategies, such as seminaries, close mentorship, or virtual simulators. Nowadays, there is a growing literature on the learning curves in urology and female pelvic floor medicine. For instance, there are some reports on operative cystoscopy for botulinum toxin bladder injections and transurethral resection of the prostate, which showed excellent safety and efficiency even at the beginning of the learning curve [9,10,11]. Similar experiences have been published regarding the surgical management of stress urinary incontinence with midurethral slings [8,12]. However, t o the best of our knowledge, there are no previous data about the urodynamics learning curve. The use of urodynamics has progressively been reduced over time. Its cost-effectiveness before prolapse surgery has been long questioned since findings rarely affect the clinical decision-making process. More recently, its routinary use before stress incontinence surgery has also been questioned [13,14]. However, these reports have led to an unjustified decrease in the use of urodynamics even for more complex indications. For instance, a recent Dutch survey revealed that almost half of urologists and gynecologists do not carry out urodynamics in complex cases, such as stress incontinence associated with large postvoid residual, poor flow, or doubts regarding the reason for incontinence [15]. This widespread reduction in the use of urodynamics poses some criticism with respect to residents’ training, in particular, in terms of underexposure to this diagnostic tool. As a consequence, defining a minimal workload necessary to adequately perpetrate skills and knowledge may be very relevant for residency programs and schools.

Our study has several strengths, including originality and prospective design. Moreover, while in most cases learning curves in urogynecology are based on the performances of a single operator, thus limiting the generalization of the data, in our series, proficiency parameters were evaluated for three different residents, making our findings more reliable [8,12]. In addition, several parameters have been considered as possible variables influenced by residents’ experience, and an adequate number of procedures for each resident has been analyzed. A limitation is the single-center design, which may limit the generalization of our findings. Thus, external confirmation would be recommended. However, these preliminary findings could be considered when organizing residency programs.

## 5. Conclusions

We found that there is a tangible learning curve for urodynamics in terms of several proficiency parameters, such as duration of procedure, perceived difficulty, call for help, accuracy of interpretation, and appropriateness of therapeutic proposal. A workload of 5 uroflowmetries and 15 cystomanometries with pressure/flow studies may be adequate to achieve a satisfactory capability to perform urodynamics.

## Figures and Tables

**Figure 1 medicina-58-00341-f001:**
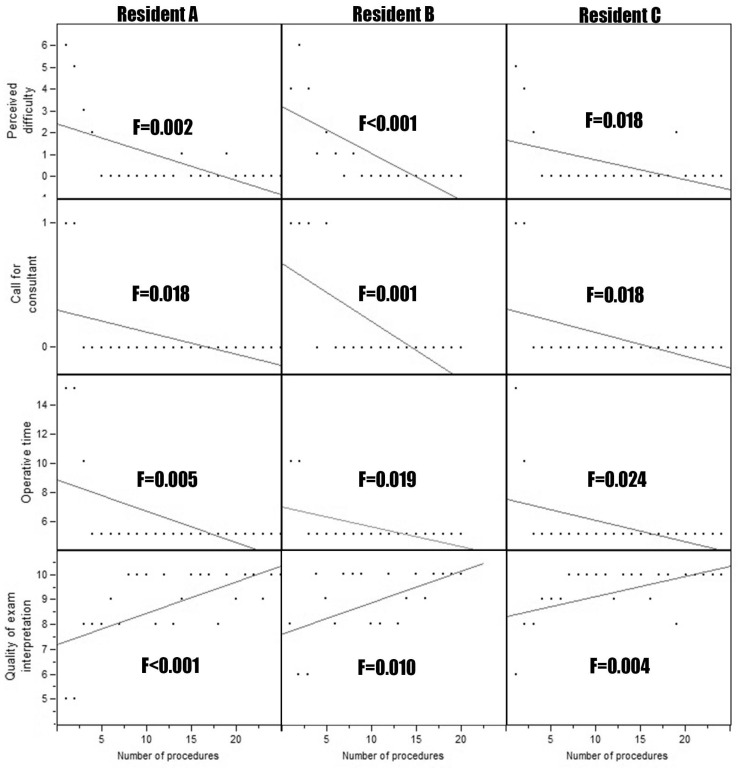
Linear regression of duration of procedure, perceived difficulty, need for consultant intervention, accuracy of interpretation for the number of procedures executed for uroflowmetry for each resident.

**Figure 2 medicina-58-00341-f002:**
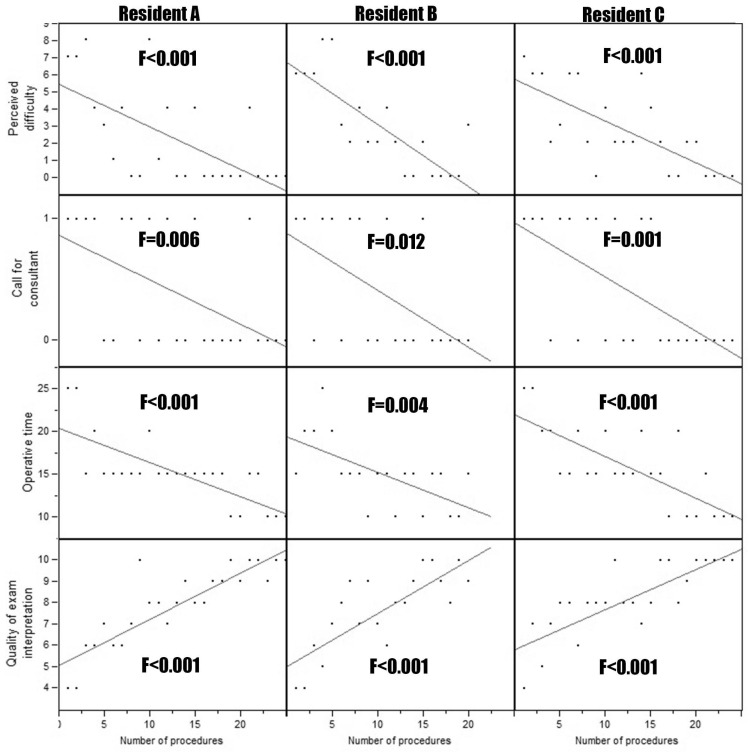
Linear regression of duration of procedure, perceived difficulty, need for consultant intervention, accuracy of interpretation for the number of procedures executed for cystomanometry with pressure/flow study for each resident.

**Figure 3 medicina-58-00341-f003:**
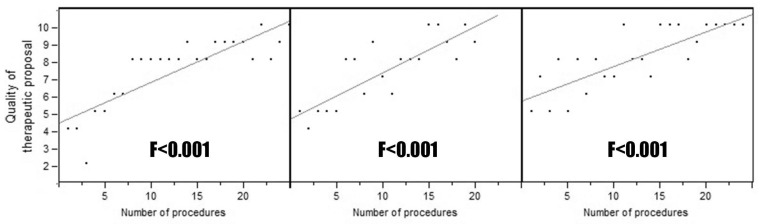
Linear regression of the appropriateness of the therapeutic proposal for each resident.

**Table 1 medicina-58-00341-t001:** Proficiency parameters for each resident analyzed in groups of 5 consecutive procedures. Data as mean ± standard deviation. The difficulty was evaluated with a 10-point VAS scale (0 = very easy, 10 = very difficult). The accuracy of interpretation of the exam was evaluated with a 10-point VAS scale (0 = very inaccurate, 10 = very accurate). The appropriateness of the therapeutic proposal was evaluated with a 10-point VAS scale (0 = very inappropriate, 10 = very appropriate).

	Group	Resident A	Resident B	Resident C
Uroflowmetry: duration of procedure (minutes)	1–5	10.0 ± 5.0	7.0 ± 2.7	8.0 ± 4.5
	6–10	5.0 ± 0.0	5.0 ± 0.0	5.0 ± 0.0
	11–15	5.0 ± 0.0	5.0 ± 0.0	5.0 ± 0.0
	16–20	5.0 ± 0.0	5.0 ± 0.0	5.0 ± 0.0
Uroflowmetry: difficulty	1–5	3.2 ± 2.4	3.4 ± 1.9	2.2 ± 2.3
	6–10	0.0 ± 0.0	0.4 ± 0.5	0.0 ± 0.0
	11–15	0.2 ± 0.4	0.0 ± 0.0	0.0 ± 0.0
	16–20	0.2 ± 0.4	0.0 ± 0.0	0.4 ± 0.9
Uroflowmetry: need for consultant intervention (n)	1–5	2	4	2
	6–10	0	0	0
	11–15	0	0	0
	16–20	0	0	0
Uroflowmetry: accuracy of interpretation	1–5	6.8 ± 1.6	7.8 ± 1.8	8.0 ± 1.2
	6–10	9.4 ± 0.9	9.2 ± 1.1	9.8 ± 0.4
	11–15	9.0 ± 1.0	9.0 ± 1.0	9.8 ± 0.4
	16–20	9.4 ± 0.9	9.8 ± 0.4	9.4 ± 0.9
Cystomanometry + pressure/flow study: duration of procedure (minutes)	1–5	20.0 ± 5.0	20.0 ± 3.5	21.0 ± 4.2
	6–10	16.0 ± 2.2	14.0 ± 2.2	17.0 ± 2.7
	11–15	15.0 ± 0.0	13.0 ± 2.7	16.0 ± 2.2
	16–20	13.0 ± 2.7	13.0 ± 2.7	13.0 ± 4.5
Cystomanometry + pressure/flow study: difficulty	1–5	5.8 ± 2.2	6.8 ± 1.1	4.8 ± 2.2
	6–10	2.6 ± 3.4	2.6 ± 0.9	3.6 ± 2.6
	11–15	1.8 ± 2.0	1.6 ± 1.7	3.2 ± 1.8
	16–20	0.0 ± 0.0	0.6 ± 1.3	1.2 ± 1.1
Cystomanometry + pressure/flow study: need for consultant intervention (n)	1–5	4	4	4
	6–10	3	2	3
	11–15	2	2	3
	16–20	0	0	0
Cystomanometry + pressure/flow study: accuracy of interpretation	1–5	5.4 ± 1.3	5.2 ± 1.3	6.2 ± 1.6
	6–10	7.4 ± 1.7	8.0 ± 1.0	7.6 ± 0.9
	11–15	8.0 ± 0.7	8.2 ± 1.5	8.2 ± 1.1
	16–20	9.0 ± 0.7	9.2 ± 0.8	9.4 ± 0.9
Appropriateness of therapeutic proposal	1–5	4.0 ± 1.2	4.8 ± 0.4	6.0 ± 1.4
	6–10	7.2 ± 1.1	7.6 ± 1.1	7.2 ± 0.8
	11–15	8.2 ± 0.4	8.0 ± 1.4	8.6 ± 1.3
	16–20	8.8 ± 0.4	9.2 ± 0.8	9.4 ± 0.9

## Data Availability

The datasets are not publicly available but are available from the corresponding author on reasonable request.

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
