# Peer review of "The Learning Curve of Urodynamics for the Evaluation of Lower Urinary Tract Symptoms"

_medicina, 2022, doi:10.3390/medicina58030341_

Round 1
Reviewer 1 Report
I have read your manuscript“The learning curve of urodynamics for the evaluation of lower urinary tract symptoms” with great pleasure. This is an interesting article. The authors suggested that 5 uroflowmetries and 15 cystomanometries with pressure/flow studies may be sufficient to complete the learning curve for the evaluation of lower urinary tract symptoms.
I have several comments:
- In “Abstract”, PFD needs to display full name.
- I am particularly concerned about the lack of cohesion between the subsequent sentences in several paragraphs of the text (e.g., page 1, lines 42-44). The author needs to be more descriptive.
- In “Table 1”, scoring units need to be added to the consultant scoring section.
- In “Figure 1”, add the colour code; why do you use red, or green?
- In “Discussion”, it is worth emphasizing the limitations of the analysis performed.
- After a duplication-check on your manuscript of antibiotics-1561883, there a few paragraphs/sentences are almost the same with the published papers (e.g., lines 94-105, 147-150, 182-184), could you please rewrite/make revisions accordingly?
Author Response
Reviewer 1
I have read your manuscript“The learning curve of urodynamics for the evaluation of lower urinary tract symptoms” with great pleasure. This is an interesting article. The authors suggested that 5 uroflowmetries and 15 cystomanometries with pressure/flow studies may be sufficient to complete the learning curve for the evaluation of lower urinary tract symptoms.
I have several comments:
- In “Abstract”, PFD needs to display full name.
Edited
- I am particularly concerned about the lack of cohesion between the subsequent sentences in several paragraphs of the text (e.g., page 1, lines 42-44). The author needs to be more descriptive.
Edited
- In “Table 1”, scoring units need to be added to the consultant scoring section.
VAS scale use no measurement units, so it was not possible to add them
- In “Figure 1”, add the colour code; why do you use red, or green?
There is no color code, we rebuild figures without colours
- In “Discussion”, it is worth emphasizing the limitations of the analysis performed.
Thank you, added
- After a duplication-check on your manuscript of antibiotics-1561883, there a few paragraphs/sentences are almost the same with the published papers (e.g., lines 94-105, 147-150, 182-184), could you please rewrite/make revisions accordingly?
We reworded the indicated paragraphs as required
Reviewer 2 Report
The authors evaluated the learning curve of residents with no previous experience to correctly perform and interpret urodynamics, and properly address and manage patient pelvic floor disorders based on urodynamics findings. They concluded that there is a tangible learning curve for urodynamics in terms of several proficiency parameters. A workload of 5 uroflowmetries and 15 cystomanometries with pressure/flow studies may be adequate to complete the learning curve.
My concerns are as follows:
- Abstract – PFD – Please provide full title before using an abbreviation.
- Introduction – The authors should present a few general sentences in regards to learning curves (if there are no specific publications regarding urodynamics) – E. g. Learning curves are complex functions and as a result contain various stages. The beginning stage is known as the initial curve, where there is generally a stepwise improvement in learning that can be applied across all medical specialties and procedures. There is a theory that states that learning occurs slower when an operator becomes more competent at a skill. This inevitably leads to reaching an expert plateau [REFERENCE: Valsamis, E.M.; Chouari, T.; O’Dowd-Booth, C.; Rogers, B.; Ricketts, D. Learning curves in surgery: Variables, analysis and applications. Postgrad. Med. J. 2018, 94, 525–530]. Some factors that may affect the rate of learning are related to the individual at hand, such as previous experience, motivation, natural talent and the ability to acquire new skills. The plateau that is inevitably reached does not necessarily indicate an expert level but instead signifies when retardation of learning has occurred. The last stage is the redirection of performance. After the plateau has been reached there is usually a slight decline in performance, which can usually be attributed to overconfidence and the ascertainment of more difficult operations [REFERENCE: Pogorelić, Z.; Huskić, D.; ÄŒohadžić, T.; Jukić, M.; Šušnjar, T. Learning Curve for Laparoscopic Repair of Pediatric Inguinal Hernia Using Percutaneous Internal Ring Suturing. Children 2021, 8, 294. https://doi.org/10.3390/children8040294].
- I do not agree with authors that Institutional Review Board (IRB) approval for this study is not mandatory. Regarding new European regulations, an IRB statement is mandatory for this prospective study.
- The authors should present clear inclusion / exclusion criteria for this study.
- Primary / secondary outcomes should be mentioned in methodology.
- It is unclear which statistical test was used to test normality of data distribution. Please add information in paragraph regarding statistical analysis.
- Characteristics of the residents should be described – years of age and experience in urodynamics.
- Table 1 – The authors should add information what presented values represent (probably mean ± SD).
- Also, how many supervisors monitored residents? Whether they had standardized criteria if there were more than one supervisor (that may be a source of bias and should be mentioned in limitations).
- For several variables (e.g. ‘Uroflowmetry: difficulty’, ‘Uroflowmetry: appropriateness of interpretation’…) the authors did not provide meaning of variables, what do these numbers represent – score? Please revise.
- The authors used the term ‘operative time’ through the text, in Tables and figures. I do not know whether this term is appropriate. Better term would be ‘duration of procedure’.
- Discussion is poorly designed. Mostly repeating the data from literature. Discussion section needs to be re-written/re-arranged. Do not present a review of literature in this section. Do not discuss your findings piecemeal. Focus on results from the main objectives of the study. Write in four sequential paragraphs (without headings); (i) summary (not data) of findings from present study; (ii) logical and coherent comparison with existing literature with focus of comparison on main objective(s); (iii) limitations of the study; and (iv) Implications for practice/policy/research with a concluding statement
- Limitations of the study have not even been mentioned.
- References should be revised according to the journal style. Also, letters in square brackets should be removed. More references should be included and discussed.
Author Response
The authors evaluated the learning curve of residents with no previous experience to correctly perform and interpret urodynamics, and properly address and manage patient pelvic floor disorders based on urodynamics findings. They concluded that there is a tangible learning curve for urodynamics in terms of several proficiency parameters. A workload of 5 uroflowmetries and 15 cystomanometries with pressure/flow studies may be adequate to complete the learning curve.
My concerns are as follows:
- Abstract – PFD – Please provide full title before using an abbreviation.
Edited
- Introduction – The authors should present a few general sentences in regards to learning curves (if there are no specific publications regarding urodynamics) – E. g. Learning curves are complex functions and as a result contain various stages. The beginning stage is known as the initial curve, where there is generally a stepwise improvement in learning that can be applied across all medical specialties and procedures. There is a theory that states that learning occurs slower when an operator becomes more competent at a skill. This inevitably leads to reaching an expert plateau [REFERENCE: Valsamis, E.M.; Chouari, T.; O’Dowd-Booth, C.; Rogers, B.; Ricketts, D. Learning curves in surgery: Variables, analysis and applications. Postgrad. Med. J. 2018, 94, 525–530]. Some factors that may affect the rate of learning are related to the individual at hand, such as previous experience, motivation, natural talent and the ability to acquire new skills. The plateau that is inevitably reached does not necessarily indicate an expert level but instead signifies when retardation of learning has occurred. The last stage is the redirection of performance. After the plateau has been reached there is usually a slight decline in performance, which can usually be attributed to overconfidence and the ascertainment of more difficult operations [REFERENCE: Pogorelić, Z.; Huskić, D.; ÄŒohadžić, T.; Jukić, M.; Šušnjar, T. Learning Curve for Laparoscopic Repair of Pediatric Inguinal Hernia Using Percutaneous Internal Ring Suturing. Children 2021, 8, 294. https://doi.org/10.3390/children8040294].
Edited as required
- I do not agree with authors that Institutional Review Board (IRB) approval for this study is not mandatory. Regarding new European regulations, an IRB statement is mandatory for this prospective study.
The following IRB statement is already written on the paper “As this was an observational analysis, and clinical management of patients was not modified by the study, it was considered exempt from Institutional Review Board (IRB) approval from the local Ethics Committee. The study was conducted in accordance with the Declaration of Helsinki. Written informed consent was obtained from all the patients before the procedure, as part of our protocol for urodynamics.”
Please note that patients signed written consent before urodynamics. Moreover, the study is about resident performance, not about patients. All residents voluntary agreed to partecipate to the study. Added to the manuscript.
- The authors should present clear inclusion / exclusion criteria for this study.
No inclusion or exclusion criteria were applied. Added in the text
- Primary / secondary outcomes should be mentioned in methodology.
Added. “As primary outcomes we evaluated the impact of experience on the following proficiency parameters: duration of procedure, perceived difficulty, need for consultant intervention, appropriateness of interpretation, and therapeutic proposal. As secondary outcome we willed to establish the workload necessary to achieve a satisfactory capability to perform urodynamics.”
- It is unclear which statistical test was used to test normality of data distribution. Please add information in paragraph regarding statistical analysis.
Since no comparison of means were applied, no normality test was necessary. Statistical analysis was made with linear regression as stated
- Characteristics of the residents should be described – years of age and experience in urodynamics.
As stated in materials and methods “None of them had previously performed urodynamics.”. Added that they were heterogeneous with respect to years of residency
- Table 1 – The authors should add information what presented values represent (probably mean ± SD).
Added
- Also, how many supervisors monitored residents? Whether they had standardized criteria if there were more than one supervisor (that may be a source of bias and should be mentioned in limitations).
There was only one supervisor (MF). Added in the manuscript
- For several variables (e.g. ‘Uroflowmetry: difficulty’, ‘Uroflowmetry: appropriateness of interpretation’…) the authors did not provide meaning of variables, what do these numbers represent – score? Please revise.
This was already specified in methods section.
-The operator was asked to evaluate each exam's perceived difficulty with a 10 point VAS scale (0=very easy, 10=very difficult).
- Appropriateness of the interpretation of the exam made by the resident was evaluated by the consultant with a 10 point VAS scale (0=very inappropriate, 10=very appropriate).
-Lastly, the resident was asked to make a hypothetical proposal of management/treatment based on their interpretation of clinical data and urodynamic findings, and this was rated by the consultant with a 10 point VAS scale (0=very inappropriate, 10=very appropriate).
- The authors used the term ‘operative time’ through the text, in Tables and figures. I do not know whether this term is appropriate. Better term would be ‘duration of procedure’.
Edited
- Discussion is poorly designed. Mostly repeating the data from literature. Discussion section needs to be re-written/re-arranged. Do not present a review of literature in this section. Do not discuss your findings piecemeal. Focus on results from the main objectives of the study. Write in four sequential paragraphs (without headings); (i) summary (not data) of findings from present study; (ii) logical and coherent comparison with existing literature with focus of comparison on main objective(s); (iii) limitations of the study; and (iv) Implications for practice/policy/research with a concluding statement
Re-arranged as required
- Limitations of the study have not even been mentioned.
Thank you, added
- References should be revised according to the journal style. Also, letters in square brackets should be removed. More references should be included and discussed
Added references and revised according to journal style
Round 2
Reviewer 1 Report
The manuscript is acceptable in its present form.
Author Response
Reviewer 1
The manuscript is acceptable in its present form
Thank you
Reviewer 2 Report
The authors performed most of the requested corrections.
- The authors were asked to add explanations for several variables in table 1 (e.g. ‘Uroflowmetry: difficulty’, ‘Uroflowmetry: appropriateness of interpretation’…) because they did not provide meaning of variables. Even though it was stated in methodology, each Table should be clear without reading the methodology section (which is standard in each international journal), so please add explanation next to each variable or in the legend of the Table.
- The Quality of English needs improvement, even I requested moderate changes, the authors did not improve English in the manuscript.
Author Response
Reviewer 2
The authors performed most of the requested corrections.
Thank you
The authors were asked to add explanations for several variables in table 1 (e.g. ‘Uroflowmetry: difficulty’, ‘Uroflowmetry: appropriateness of interpretation’…) because they did not provide meaning of variables. Even though it was stated in methodology, each Table should be clear without reading the methodology section (which is standard in each international journal), so please add explanation next to each variable or in the legend of the Table.
The Quality of English needs improvement, even I requested moderate changes, the authors did not improve English in the manuscript.
We added explanations for table 1.
We now passed the manuscript through an English editing service